# Differential Immune Response Patterns Induced by Anionic and Cationic Lipid Adjuvants in Intranasal Anti-Influenza Immunization

**DOI:** 10.3390/vaccines12030320

**Published:** 2024-03-18

**Authors:** Anirban Sengupta, Noha Al-Otaibi, Claudia Devito, Francisca Lottersberger, Jorma Hinkula

**Affiliations:** 1Centre for Infectious Medicine, Department of Medicine Huddinge, Karolinska Institutet, 141 52 Stockholm, Sweden; anirban.sengupta@ki.se; 2King Abdulaziz City for Science and Technology (KACST), Riyadh 12354, Saudi Arabia; naalotaibi@icloud.com; 3Saudi Drug and Food Authority (SFDA), Riyadh 13513, Saudi Arabia; 4HD Immunity, 126 25 Nacka, Sweden; succinumab@telia.com; 5Department of Biomedical and Clinical Sciences, Linköping University, 581 83 Linköping, Sweden; francisca.lottersberger@liu.se

**Keywords:** adaptive immunity, Treg Th17 balance, cell-mediated immunity, adjuvant influenza, dendritic cell response, lymphocyte function, humoral response

## Abstract

Currently, vaccine development against different respiratory diseases is at its peak. It is of utmost importance to find suitajble adjuvants that can increase the potency of the vaccine candidates. This study aimed to determine the systemic and splenic immune mechanisms in mice models induced by anionic and cationic lipid adjuvants in the presence of the vaccine-candidate influenza antigen hemagglutinin (HA). In the presence of the HA antigen, the cationic adjuvant (N3) increased conventional dendritic cell 1 (cDC1) abundance with enhanced MHCI and CD80-CD86 costimulatory marker expression, and significantly higher CD8T and Th17 populations with enhanced interferon-gamma (IFNγ) expression in CD8T and CD4T populations. Conversely, the anionic adjuvant (L3) increased the cDC2 population percentage with significantly higher MHCII and DEC205 expression, along with an increase in the CD4T and regulatory T cell populations. The L3-treated group also exhibited higher percentages of activated B and plasma cell populations with significantly higher antigen-specific IgG and IgA titer and virus neutralization potential. While the anionic adjuvant induced significantly higher humoral responses than the cationic adjuvant, the latter influenced a significantly higher Th1/Th17 response. For customized vaccine development, it is beneficial to have alternative adjuvants that can generate differential immune responses with the same vaccine candidate antigen. This study will aid the selection of adjuvants based on their charges to improve specific immune response arms in the future development of vaccine formulation.

## 1. Introduction

Recently, advancements in vaccine research have shown promising results in curbing many epidemics and pandemics, even preventing some from happening completely [1]. In the last few decades, vaccine formulation and design have been moving towards better and more promising targets, with better safety and efficacy. The choice and use of adjuvants are paramount in developing an effective vaccine [2]. They are particularly important for the development of the influenza vaccine, where there is a need to optimize the immune response [2,3]. Individuals with suppressed or weaker immune responses, such as the pediatric and elderly populations, depend largely on the adjuvants for vaccine efficacy [4]. Hence, the proper selection and administration of adjuvants and the vaccine antigen are crucial.

The primary immune barrier at the initial infection site, the mucosa, holds great value in preventing the entry of pathogenic microorganisms. According to the World Health Organization (WHO), respiratory tract viral infections are among the top 10 causes of death [5]. Any type of preventive treatment at mucosal sites would most likely be beneficial. Thus, viral vaccines given at mucosal sites can mimic the natural entry pathway of infectious agents by providing local mucosal immunity, resulting in the strengthening of systemic immunity [6]. Often, mucosal-administered vaccines contain pure antigens, which may require support from adjuvants to enhance their passage through the mucosal barrier, thereby reducing the amount of antigen needed to induce immune response [7]. At the same time, it is also important to prevent the tolerogenic response elicited by the mucosal sites against pathogens and vaccine candidates [8]. Adjuvants and antigens given at the mucosal site may cause local irritation or be flushed out before reaching immune presentation, resulting in a weaker or no immune response [9,10]. Therefore, adjuvants that have similarities to natural physiological mucosal substances, such as endogenous mucosal lipids or fatty acids, may be attractive adjuvant candidates as they may be better tolerated whilst still inducing immune responses [9,10].

The properties of these adjuvants (anionic L3 and cationic N3) have been reported in previously published patents [11,12,13,14] and clinical trials [15,16]. Briefly, they are a combination of a monoglyceride and a fatty acid that can stimulate the immune system to produce antibodies and induce protective immunity. They form a lipid-based dispersion with particles of less than 100 nm [17]. The general composition of the emulsion includes the following: (i) Monoglycerides, selected from a group with the general formula of I-acyl-glyceride, wherein the number of carbon atoms in the acyl chain varies between 8 and 24, typically between 12 and 18; the acyl chain is either saturated or unsaturated. (ii) One or more fatty acids comprising 6 to 24 carbon atoms, wherein the acyl chain contains one or more unsaturated bonds. Both the monoglycerides and the fatty acid concentrations are in the range of 0.1–50 g per 100 mL formulation, typically in the range of 1–20 g per 100 mL water. When monoglycerides and fatty acids are formulated together, the percent ratio of monoglyceride to fatty acid varies between 1 and 99%, preferably between 10 and 90%. (iii) Water [11,12,13]. The osmolarity is close to physiological levels, ranging from 290 to about 310 mOsm/kg. The surfactant is a hydrophilic, inert, and biocompatible surfactant, such as a poloxamer, Pluronic F68, or Pluronic 12714.

Previous studies from our lab and others have proved the potential and safety of the anionic L3 adjuvant in pre-clinical settings for intranasal administration of influenza vaccines [18,19,20,21]. Clinical score, histopathological studies, virology load, physical parameters, body weight loss, electron microscopic study of the nasal mucosa, micromorphological alterations in the nasal cavity or the olfactory bulb, lung morphology, and histology all indicated the safety of the adjuvants in preclinical studies with rats [12]. The toxicity assessment for establishing the safety of these adjuvants in the above parameters was also performed on ferrets [11,14,20]. Two multi-centered, placebo-controlled, partially blinded clinical trials (NCT03437304 and NCT02998996) performed on a total of 460 human participants confirmed the safety of these adjuvants for use in humans [15,16]. No LPS or endotoxin-mediated immunotoxicity was reported in any of these studies.

These adjuvants have been reported before to function better than the other mucosal adjuvants, such as Cholera toxin and alum [22]. The use of the adjuvants in an inactivated influenza vaccination strategy was at least equivalent to the GSK monovalent pandemic vaccine (GSK H1N1), Novartis trivalent inactivated vaccine (Novartis TIV), and GSK trivalent inactivated vaccine (GSKTIV)15. The adjuvants increased the immune responses in both young and elderly populations, as observed in animal studies [15,16,18,19]. The adjuvants have been evaluated in HIV, diphtheria, and tuberculosis vaccines [23,24]. L3 adjuvant administration in influenza vaccination reduced the viral load in the lungs and induced higher anti-viral antibody titers in mice and ferrets [14,20], as well as enhanced neutralization potential [18]. The cationic version of the same adjuvant (N3) was first used by our lab in 2006 for HIV vaccination [25], and was later also studied in influenza [18] and SARS-CoV-2 vaccination [26].

The efficacy of the lipid adjuvants in the mucosal surface and the local immune response elicited following intranasal administration have been explored, reported, and reviewed previously [19,27,28,29]. However, the systemic and splenic immunity elicited by differentially charged lipid adjuvants remains largely unexplored. This is essential for future vaccine design to better understand the holistic immune response induced by these adjuvants at the systemic level. Previous studies have provided information regarding the role of these adjuvants in virus-challenge experiments with influenza [18,19,20,21]. Here, we focused on investigating how these adjuvant-based vaccines prepare the host immune system against influenza before being challenged in a mice model. The mice study groups are presented in Table 1. This study aimed to reveal the mechanism of action of these adjuvants (but only the adjuvant–antigen vaccine formulation without the influence of the whole virus challenge) in influencing the maturation, activation, and differentiation of the major immune cells in the spleen and also to investigate the systemic humoral immune response. The novelty of our study is that we identified the differences in the action of differentially charged adjuvants in boosting specific immune pathways.

## 2. Materials and Methods

### 2.1. Animal Use and Experimental Groups

Female, 6–8 weeks old C57Bl/6 mice were obtained from Janevier, Sollentuna, Sweden. The mice were assigned to seven experimental groups (*n* = 5 per group), including one control group. The control group was given a saline solution equal to the amount of treatment administered to the experimental groups. Human endogenous adjuvant L3 (anionic) and a derivative of N3 (cationic) lipid formulations were obtained from the IP rights-holders Eurocine Vaccines AB, Solna, Sweden. HA proteins were purchased from SinoBiological (Cat. No. 11085-V08H, Sinobiologicals, Eschborn, Germany). The formulations administered to each study group are provided in Table 1 and described in Section 2.2. This study was carried out in strict accordance with the guidelines in the Guide for the Care of Laboratory Animals at Linköping University. The protocol was approved by the Committee on Ethics of Animal Experiments of Linköping University (Protocol Numbers Dnr 18053-2020 and Dnr 00234-2022). The number of animals used was kept to a minimum in accordance with the 3R recommendations whilst still allowing statistical comparison between study groups. The duration of the experiment was 42 days from the first dose of the vaccine. No animals were found dead in the cage throughout the study period. 

Animal health and behavior were monitored thrice daily from the start of the study with about 8 h intervals during the first 10 days, then daily thereafter until the study ended. Body weight was measured daily for all mice in the study. To reduce suffering and distress, animals were kept in ventilated Makrolon cages with 2–5 animals/cage enriched to increase the variety of stimulant materials [30], such as bedding and nesting materials, deeper bedding materials, tunnels/shelter, paper, and cotton pads, with water and food pellets ad libitum [31]. All cages were kept at ambient temperature (22–24 °C) and in a light/dark cycle condition according to the normal 24 h cycle.

All animal sacrifices were performed on animals sedated with isoflurane (4%/air) (*n* = 35, Table 1), and efforts were made to minimize suffering. The anesthetic induction chamber was filled with 4% isoflurane in oxygen at a flow rate of 1 L/min. Humane endpoints were used according to some of the recommendations of Nuno et al., 2012 [32]. During the isoflurane anesthesia, a humane endpoint was established to minimize potential suffering. This involved monitoring the depth of anesthesia by assessing pedal reflex, respiratory rate, and the absence of a response to toe pinching. Once a deep anesthesia was achieved, subcutaneous administration of analgesic (Buprenorphine, Merck, Darmstadt, Germany: B-044-1ML) was provided to alleviate suffering and pain. The analgesic dose was determined based on the individual weight of the mice according to the manufacturer’s instructions. Following the analgesic administration, euthanasia was performed by cervical dislocation. This method is widely accepted as a rapid and humane method to induce death. After sacrifice, death was ensured by assessing the absence of vital signs, including heartbeat and respiration. At the end of the study period, blood samples and the spleens of all animals (*n* = 35) were harvested for immunological analyses at the laboratory.

### 2.2. Vaccine and Adjuvant Formulation and Administration Procedure

The L3 and N3 lipid adjuvant formulations had a 2% (*w*/*v*) final concentration by mixing with either 0.01 µg, 0.1 µg, or 1 µg of HA recombinant protein to a 5 µL volume/mouse. The vaccine formula was prepared with a 1:1 (*v*/*v*) HA and the N3 emulsion to achieve a final concentration of 2% N3 lipid formulation. The same adjuvant formulations of L3 and N3 were also used and characterized in our recently published study on SARS-CoV-2 vaccination [26], and also in previous influenza [18] and HIV [25,33] vaccination studies from our lab. There is always a possibility that the LPS or endotoxin may influence the outcome of the immunological analysis with the intranasal delivery. However, no adverse LPS/endotoxin-mediated effects on the immune response have been reported in the previous pre-clinical [11,12,17,18,19,21,25,26] and clinical trials [15,16].

In this study, we evaluated two differentially charged adjuvants that we received from Eurocine Vaccines AB. The negatively charged adjuvant (L3) is prepared by mixing oleic acid (0.46 g) and lauric acid (0.34 g), which is subsequently sonicated with 9.2 mL of 0.1 M Tris buffer (pH 8.0). The pH of the final solution is adjusted to pH 8.0 with 5 M NaOH. The final concentration of the lipid formulation is 2% [12,14]. The positively charged adjuvant is prepared by mixing mono-olein (0.15 g), oleic acid (0.12 g), and squalene (0.53 g). The mixture is subsequently sonicated with 0.2 mL of 0.1 M Tris buffer (pH 8.0). The final formulation is adjusted to pH 8 with 5 M NaOH. The final concentration of the lipid formulation is 2% [11,12,14].

The key impacts that can be induced by LPS/endotoxin in the system are inflammation with a major impact on immune cells, and systemic level aberrations in blood pressure and heart rate, etc. A blood sampling study for hematology, clinical chemistry, and immunological parameters, as well as ECGs performed to study cardiac function and urine sampling for human participants, indicated no detrimental effects due to LPS/endotoxin contamination [15,16].

The sedation procedure was performed as previously described [34,35]. Each time, a single mouse was taken out of the chamber, held in a supine position, then given a nasal drop of 5 μL/nostril antigen–adjuvant formulation, using a 1–10 µL pipette, and placed back inside the chamber in a supine position. The immune stimulant solution was administered as a single dose on day 0 and day 21 [34,35]. Serum samples for serological studies were collected on the 7th, 14th, 28th, and 42nd day post-first vaccine dose. For the first three timepoints, 5–10 µL blood was drawn from the tail vein. The remaining blood and spleen were harvested on the day of sacrifice, the 42nd day post-first immunization dose (i.e., 21st day post-second immunization dose).

### 2.3. Flow Cytometry

Flow cytometry was performed on the splenic single-cell suspension, as described previously [36,37]. Mice were sacrificed on the 21st day post-second immunization dose, and spleens were harvested. Single-cell suspensions of the splenocytes were prepared by mechanical tapping and gentle circular motion with the flat end of the piston/plunger from a sterile 3cc syringe on the minced spleen. This released the splenocytes, which were then passed through cell strainers. The cells were washed with phosphate-buffered saline (PBS) and centrifuged at 1000 rpm for 10 min at 4 °C, and the supernatant was discarded. Pellets were resuspended in a 2 mL red blood cell (RBC) lysis buffer (Miltenyi Biotec, Lund, Sweden 130-094-183) to remove the RBCs. They were incubated for 5 min at room temperature. After 5 min, the cells were centrifuged at 1000 rpm for 10 min at 4 °C, and the pellets were resuspended in PBS for washing. The washing steps were repeated three times. The cells were resuspended in cell staining buffer (CSB) (Biolegend, Nordic Biosite AB, Sweden: 420201) containing 2% fetal bovine serum (FBS) (Merck: F7524) in PBS. Anti-CD16/32 antibodies (Miltenyi Biotec:130-092-574, Biolegend: 101301) were used to block the non-specific binding sites in the cells and enhance the specificity of binding by the fluorochrome-tagged target antibodies. They were then incubated with fluorochrome-conjugated antibodies (Miltenyi Biotec), following the manufacturer’s protocol for antibody dilution, incubation duration, etc. Table 2 describes all the antibodies employed for the identification of the cellular phenotypes and their activation state. The antibody combinations for individual phenotypes or activation status are well established and have been reported before [38,39,40,41,42,43]. Forward scatter (FSC) versus side scatter (SSC) gate reference is used to determine the viable cell proportion. Isotype-matched antibodies (Milteny Biotec) were used as staining controls. Data compensation for the fluorescence spillover to other filters was performed using voltage and software gating. Fluorescence signals from the labeled cells were measured using a BD AriaIII Machine (BD Biosciences, Sweden) and analyzed using FlowJo software version 10.5.2. The same gating strategies, with the same set of antibodies and machine software settings, were used in our recently published work [44,45].

### 2.4. Serology Screening

Blood samples (1.5 mL) were collected from the experimental animals on the day of sacrifice. In addition, 5–10 µL of blood was drawn from the tail veins on days 7 and 14 post-first dose, and on day 7 after the second dose (i.e., 28th day after the first dose). The serum (0.15 mL) was separated following the standard procedure. For serum screening, 96-well microplates (Nunclon, Copenhagen, Denmark) were coated with 0.5 µg/mL recombinant HA proteins in sterile PBS and incubated overnight at 4 °C. Serum samples were diluted in PBS with 2.5% skim milk buffer with 0.05% Tween 20. Serial dilutions from 1:100 to 1:100,000 were prepared, and 100 µL from each dilution was transferred into duplicate wells in antigen-coated plates, followed by incubation at 37 °C for 90 min. Thereafter, the plates were rinsed with PBS+ with 0.05% Tween 20 (PBS-T). Horseradish peroxidase (HRP)-labeled conjugate goat anti-mouse IgG (BioRad #1706516, Richmond, CA, USA) and IgA (BioRad: STAR137P) were added to each well (100 µL/well) and incubated at 37 °C for 90 min. Then, the plates were washed again with PBS-T, followed by the addition of 0.2 g o-phenylenediamine (OPD) (Thermo Fisher Scientific, Sweden: 34006) to 0.03% H_2_O_2_ in 1 mL. Then, the plates were kept in the dark at room temperature (RT) for 30 min. The reaction was terminated by adding 100 µL of 2.5% H_2_SO_4_ to each well, and the absorbance was measured at OD490. The cutoff value for positive reactivity was calculated from the mean OD490 plus SD for negative control samples.

### 2.5. Hemagglutination Inhibition Assay

The hemagglutination inhibition assay (HIA) was used to evaluate the presence of neutralizing anti-HA antibodies against viral influenza A in the serum of individual mice. An HIA titer ≥ 40 was defined as a protective amount of serum antibodies. Briefly, sera from individual mice were treated with receptor-destroying enzyme (RDE) (Deben Diagnostics, United Kingdom: Cat No. 370013) overnight at 37 °C to remove non-specific serum HAI inhibitors. RDE was inactivated by incubation at 56 °C for 30 min, followed by the addition of 350 µL of 0.9% NaCl. The HI assay was initiated by adding 25 µL PBS to each well of a microtitre plate, followed by the addition of 50 µL of RDE-treated serum. The serum was diluted in two-fold serial dilutions. Then, 25 µL of influenza A/California/07/2009(H1N1)pdm containing four hemagglutinating units (HU) was added to each well. The plate was shaken, covered, and incubated at 20–25 °C for 15 min. Subsequently, 50 µL chicken erythrocytes was added and mixed, followed by incubation for 1 h at 4 °C. Thereafter, the plate was evaluated for hemagglutination and the degree of HIA.

The formula used to calculate the hemagglutination inhibition titer is:HI Titer = Last Dilution of Serum/Hemagglutination Unit (HAU) of Antigen Titer 

The last dilution of serum that shows complete inhibition of hemagglutination is considered the HI titer, which represents the reciprocal of the serum dilution. This titer indicates the amount of neutralizing antibodies present in the serum sample against the specific influenza strain used in the assay. The titers from each group of mice were subjected to analysis to find the geometric mean and the geometric SD values, which were plotted in a graph. The HI assay titers were plotted on a logarithmic scale to visualize the relationship between the serum dilutions and the corresponding inhibitory titers. Logarithmic transformation was applied to the dilution factors (y-axis) to accommodate the wide range of dilutions used in the assay.
Log Dilution Factor = Log10 (1/Dilution Factor)

### 2.6. Statistical Analyses

Data analysis and data representation were performed using GraphPad Prism Version 9.4.1. Comparisons between study groups were performed with non-parametric methods using the Mann–Whitney U test. Corrections for the multiple comparisons were made during statistical analysis of the data. *p* < 0.05 was considered statistically significant.

## 3. Results

### 3.1. Anionic (L3) Adjuvant-Treated Group Showed Enhanced Activation of cDC2 While Cationic (N3) Adjuvant-Treated Group Facilitated cDC1 Population Activation

Two major subsets of conventional dendritic cells, B220-MHCII+CD11c+CD11b-CD8a+ cDC1 and B220-MHCII+CD11c+CD11b+CD8a- cDC2, were gated out from the live single cell populations (Figure 1A). While cDC1 plays a major role in the antigen presentation to cytotoxic T cells, cDC2 is involved in the activation of CD4+ T cells [45,46]. The population percentage of cDC1 showed no significant changes during the L3 treatment compared to that of the control, whereas a significant increase in cDC1 was observed post-N3 treatment in comparison to that in the control (Figure 1B). On the other hand, L3 treatment did increase the cDC2 population with its highest dose, i.e., L3I, with no significant changes in the cDC2 population percentage in N3 treatment as compared with that of the control (Figure 1B).

L3 treatment also increased CD80+CD86+ cDC2 populations (Figure 1C), MHCI expression on cDC2 (Figure 1D), and DEC205 expression within cDC2 (Figure 1E) compared to those of the control cDC2 and N3-treated cDC2 cells. On the other hand, N3 treatment resulted in a significantly higher CD80+CD86+ cDC1 population (Figure 1C), remarkably high MHCI+ expression on cDC1 (Figure 1D), and higher DEC205 expression within cDC1 (Figure 1E). We also observed a difference in the MFI level expression of DEC205 and MHCI, which signifies that although almost all cDC1 and cDC2 expressed them in all the groups, there was a difference in the expression level of these proteins (Figure 1D,E).

These findings are significant because they demonstrate a differential response to differentially charged adjuvants. This suggests that while anionic lipid adjuvants caused an increase in the cDC2 population percentage with better co-stimulation and activation, cationic lipid adjuvants tend to facilitate the same for the cDC1 population.

### 3.2. Increase in Cytotoxic CD8T Cells and Costimulatory Marker CD28 Expression in the N3 Treated Groups Observed

CD3+ cells were gated out from the live single-cell population, and the population percentages of CD3+CD4+ and CD3+CD8+ T cells were estimated (Figure 2A). CD3+CD4+ T cells were also subjected to forkhead box P3 (FOXP3) and RAR-related orphan receptor gamma (ROR*γ*T) staining to determine the regulatory T cell (Treg) and Th17 cell populations (Figure 2A). Cytotoxic CD3+CD8+ T cells are indispensable for cell-mediated immunity. The population percentage of the CD8T cells was higher when the N3 adjuvant was used, even with lower doses of antigens (Figure 2B). The absolute number of CD3+CD8+ T cells was determined for better comparison (Figure 2C) and showed the same trend. The MFI calculated for CD28 expression in cells stained with CD3+CD8+CD28+ was stronger, representing their higher expression in the N3 group (Figure 2D).

A significantly higher CD8T cell population with enhanced CD28 expression, along with the higher MHCI expression and activation of cDC1 in the N3-treated group, indicates upregulation of cDC1- CD8T activation in post-cationic adjuvant-treated mice.

### 3.3. Higher Population Percentage of CD4T Cells in the Anionic Adjuvant Treatment with Altered Skew of the Treg:Th17 Ratio

The population percentage of the CD4T cells significantly increased in both the anionic L3I and cationic N3I groups, as compared to the control (Figure 2D). The absolute number of CD3+CD4+ T cells was determined for better comparison (Figure 2E), which also showed the same trend. Interestingly, in our investigation, we found that the expression of RORγT and FOXP3 was significant in CD3+CD4+ T cells. Regulatory T cells express FOXP3, while more immunogenic Th17 cells express RORγT [47]. The CD3+CD4+FOXP3+ Treg cell population was significantly higher in the anionic L3 groups (Figure 3A). In contrast, the CD3+CD4+RORγT+ Th17 cell population was of a higher percentage in the cationic N3 group (Figure 3B). Even the lowest dose, N3III, induced a significantly higher Th17 cell population compared to the non-adjuvant control (Figure 3B).

Treg cells are tolerogenic, while Th17 induces a stronger immunogenic response and drives the Th1 type of immune response [37,47]. Imbalances between Treg and Th17 have been reported in multiple diseases [47,48,49]. A low Treg:Th17 ratio in anionic N3 groups was observed as compared to the L3 groups (Figure 3C), indicating a proinflammatory Th1/Th17 response in cationic-treated groups. 

These findings are significant as we can conclude that, despite both the adjuvant groups sharing an almost similar CD4T population percentage, cationic N3 activates more Th17 subsets than Tregs, indicating its potential for inducing an immunogenic and proinflammatory response.

### 3.4. N3 Treatment Induces Significantly More Interferon-Gamma (IFNγ)-Producing T Cell Subsets

The CD3+CD4+ helper T cells and CD3+CD8+ cytotoxic T cells from the spleen were permeabilized and stained for IFNγ expression. Both L3 and N3 treatments enhanced the expression of IFNγ in comparison to that in the controls. However, the expression intensity, as observed by MFI, was stronger in the CD8T cells, especially post-N3 treatment (Figure 3D). However, analysis of the IFNγ+ CD4T and IFNγ+ CD8T cell populations revealed that the absolute number was significantly high for the CD4T cells in both the L3 and N3 treatments compared to that in the control group (Figure 3E). These data, along with the previous findings of enhanced T cell immunity via enzyme-linked immunosorbent spot (ELISPOT) assay post-N3 treatment [18], indicate a dependency on cell-mediated immunity in the cationic-adjuvant-treated mice.

### 3.5. Significantly Fewer Immature B Cells and More Activated B and Plasma Cells in the L3-Treated Group Than in the N3 Groups Indicated an Elevated B Cell Response in Anionic-Adjuvant-Treated Mice

Live single-cell splenic suspensions from each group were gated to obtain different B cell subsets [42,43]. After identification of B220+ and B220^low^ cell populations (Figure 4A), we obtained the populations of B220/CD45R+CD19+CD27+MHCII+CD40+CD80+ memory B cells (Figure 4B), B220/CD45R+CD19+IgM^high^+IgD^low/neg^+CD43− immature/transitional B cells (Figure 4C), B220/CD45R+CD19+IgM+IgD+MHCII+CD138− activated B cells (Figure 4C), and B220^low^ CD19 ^low/neg^ IgM−IgD−CD138+ long-lived plasma cells (Figure 4D).

A significant drop in immature B cells compared to the non-adjuvant control group was observed in both the L3 and N3 treatment groups (Figure 4E). The N3I treatment showed a significantly higher level of immature B cells compared to the L3I-treated group (Figure 4E). Significantly higher population percentages of memory B cells were observed in the L3I-treated group compared to the N3I and control groups (Figure 4E). L3 treatment also induced the activation of B cells, as well as the plasma cell population percentage compared to that in N3-treated groups (Figure 4F).

Significantly more activated B cells and plasma cells and fewer immature B cells were observed in the L3 treatment compared to the N3 treatment, indicating an enhanced humoral response post-anionic L3 adjuvant treatment.

### 3.6. Serum IgG Levels against Influenza HA Were Significantly High in Both the L3I and N3I Groups, with Significantly Higher IgA Levels and Virus Neutralization Capacity in the L3 Group

To assess the humoral response induced by these two adjuvants, we performed ELISA to investigate the IgG and IgA antibody titers for the HA antigen of influenza. As shown in Figure 5A, both the adjuvants induced a significantly higher titer of anti-HA IgG compared to that of the non-adjuvant control at the same antigen dose. In a comparison of the two adjuvant groups, the L3 groups (L3I and L3II) showed significantly higher antibody expression than the N3 groups (Figure 5A). We evaluated the anti-HA IgG expression of all the mice groups on days 7, 14, 28, and 42. Significantly higher anti-HA IgG expression was observed in the L3 group after the first dose and stable expression throughout the study period. However, there was a sudden and significant increase in the expression in the N3 groups post-second dose. It was also observed that the lower dose of antigen in L3II induced significantly higher expression than N3II (Figure 5A). Anti-HA IgA responses were critically low in all the N3-treated groups (Figure 5B). Thus, only the L3 treatment can induce anti-HA IgA, which was significantly higher in both the N3-treated and non-adjuvant groups.

While studying the viral neutralization capacity of these serums against the influenza virus, we observed significantly higher HI assay titers in both the L3I and N3I groups compared to that of the non-adjuvant control, with the L3I group exhibiting a significantly higher response than that of the N3I group (Figure 5C). Even the lower dose of antigen in the L3II group provided seroprotection above the cutoff value of 40, while it was below the protective level in the N3II group (Figure 5C).

Thus, we conclude that the L3 treatment exhibited a significantly higher humoral response with more immunoglobulin synthesis and neutralization potential than the N3 groups.

## 4. Discussion

Novel strategies are constantly being developed to combat viruses and other pathogens. The search for the perfect adjuvant with the potential to optimize the immune response of the host is critical in this area. In this study, we identified the mechanistic details of the functional alteration in key immune cells and molecules in shaping the immune response against influenza induced by differently charged lipid adjuvants. While both the anionic L3 and cationic N3 adjuvants induced remarkably enhanced immune responses compared to that of the non-adjuvant control, each had a preferential immune pathway for achieving this.

Anionic L3 adjuvants could induce higher cDC2 populations (Figure 1B) with better CD80/86 expression (Figure 1C) and DEC205 levels (Figure 1E). DEC205 is a critical dendritic cell protein marker in antigen uptake, as it is recycled through MHCII-rich late endosomal compartments, thereby increasing antigen presentation to CD4+ helper T cells [50,51]. An increase in DEC205 expression in the L3 group compared to that in the N3 groups (Figure 1E) justifies the elevated CD4T-cell-mediated B cell activation and enhanced humoral response that was observed in the L3 groups.

Increases in the cDC1 population percentage (Figure 1B) with enhanced MHCI expression (Figure 1D), significantly higher Th17 population (Figure 3B), and higher CD8T population and CD28 expression (Figure 2B–D), as well as increased Th1 cytokine IFNγ expression (Figure 3D,E) in the N3-treated group indicated a Th1/Th17 dominant response induced by the cationic lipid adjuvant. With higher intracellular levels of IFNγ within both the CD4T and CD8T cells in the N3 groups, it is evident that the N3 adjuvant promotes a considerably higher cell-mediated immune response compared to that of the L3 treatment (Figure 3E,F). This supports our previous finding via ELISPOT assay against the influenza antigen, where the N3-treated group produced more IFNγ spots than the L3 groups [18]. While the L3 group showed a 7-fold increase in T cell response as pg/mL of IFNγ over non-adjuvanted formulation, a 60-fold increase was reported in the N3 group [12]. It should be noted that both the L3 and N3-treated groups induced a considerable level of IFNγ compared to that in the non-adjuvant control group.

We should clarify that in this study, we are not proposing that either of these adjuvants is better than the other. Treatment with either L3 or N3 does not lead to ‘all or nothing’ activation of either the Th1/Th17 response or the Th2 humoral response. Both adjuvant subtypes are potent enough to activate both arms. Both the L3 and the N3 treatment showed enhanced immune responses compared to the non-adjuvant antigen control. The main premise of this study was to highlight that L3 preferred the humoral arm while N3 induced stronger Th1/Th17 immunity. Both are effective in activating and enhancing their non-preferred arm and have the potential to provide better immune protection. This knowledge of the biases of activation of the immune responses can be used to devise customized vaccine formulations for groups of individuals with inherent suboptimal responses from either humoral or cellular immunity. At the same time, it is well-known that many vaccine-candidate antigens fail to activate either humoral or cellular immunity. Administration of those antigens, along with suitable adjuvants with the ability to elicit the response from their weaker arm, will be very beneficial in providing better protection. We provide a comparative summary of the immune responses elicited by these adjuvants in Table 3.

The anti-influenza HA-specific serum IgG titer was high in both adjuvant treatment groups compared to that in the control (Figure 5A). In combination with the significantly low immature B cell and high plasma cell populations in the L3 group (Figure 4F), this indicates a stronger humoral response induced by the L3 adjuvant. L3 treatment also induces significant enhancement in HA-specific IgA titer compared to that of the non-adjuvant control, while no significant increase was observed in the N3 groups (Figure 5B). As reported previously, N3 vaccinations fail to stimulate the synthesis of influenza-specific serum IgA [18,52,53], although they have the potential to secrete mucosal IgA specific to the pathogen [18]. This difference may be due to the differential source of IgA. While the mucosal IgA is produced from the nasal-associated lymphoid tissue (NALT), the serum IgA originates mainly from the plasma cells of the spleen and the lymph nodes [54]. This fact justifies the higher splenic plasma cell level in the L3 treatment than that in the N3 treatment (Figure 4F), as the L3 treatment produces both anti-influenza IgG and IgA, while the N3 treatment mostly induces IgG production only (Figure 5A,B).

Previously, serum collected after each dose of immunization before the viral challenge with these adjuvants showed a steady increase in the HA-specific antibody level with enhanced neutralization potential [18]. The neutralization potential, determined by HI titer, was significantly higher in both the adjuvant-treated groups compared to that of the control group (Figure 5C). The titer was significantly higher in the L3 treatment compared to that in the N3 treatment, as shown in Figure 5C and a previous study [18]. The CDC report contains the statement, “…serum HI antibody titers of 40 are associated with at least a 50% reduction in risk for influenza infection or disease in populations” [55]. Higher titers provide significantly higher risk reduction of the infection and, thus, better protection. The neutralization potential determined by the HIA titer of both higher antigenic doses (L3I and N3I) showed seroprotection levels of HIA titer > 40. However, the protective titer was significantly higher in the L3I group than in the N3I, indicating its potency for better humoral protection. We also found that at a lower dose of antigen, L3II was induced significantly more than N3II. The seroprotection level of L3II was above the cutoff value of titer > 40, while that of N3II was not (Figure 5C). This is another key finding from this study as we investigated the effect of the lowering of the dose of the antigen in the vaccination formulations. High doses of the antigens in the vaccines have been associated with detrimental side effects [56]. The use of both of these adjuvants showed promising immune responses in some of the parameters studied here, even with low antigenic doses. Earlier studies with these adjuvants showed a significant reduction in the influenza virus RNA copy number in the lungs [21], even with a low antigenic dose.

Since we focused on identifying the immune response exhibited by the administration of an adjuvant–antigen vaccine formulation alone and not from the whole virus, we did not assess the post-challenge status in this study. It has already been reported that these adjuvants proved to be effective in eliciting an immune response post-viral challenge in influenza [18,19,20,21]. Thus, we limited the scope to study one anionic and one cationic variant of the lipid adjuvants here. It will be interesting to explore other variants of lipid adjuvants in future studies. Another limitation of the study is that we did not study the long-term or time-dependent changes in the immune response and focused only on one timepoint. This dynamic change in the immune response might also shed some light on the immune profile generated by these adjuvants.

The charge of the vaccine formulation (antigen + adjuvants) is important for its efficacy in boosting immunity. It affects the rate and amount of absorption of these vaccine formulations to the immune cells. It also influences the delivery and drainage of the formulations into the afferent lymphatics by passive diffusion and subsequently enables them to enter the lymph nodes [57]. Net negative charge allows an optimum delivery system due to the collagen fibers and negatively charged glycosaminoglycans 107 that make up the interstitial matrix of the afferent lymphatic vessels [58]. One commonly used natural polymer adjuvant, chitosan, has a high cationic charge, which helps it to form a tight complex with anionic nucleic acids [59]. Aluminum has been used for over a hundred years as an adjuvant. Its hydroxide variant (aluminum oxyhydride) binds to negatively charged proteins efficiently, whereas the phosphate variant (aluminum hydroxyphosphate) binds to positively charged proteins [60]. For nanoparticle-based vaccines, cationic charge facilitates their retention on the injection side, dendritic cell activation, and induction of Th1 responses [61]. The exact underlying mechanisms that drive the immune response by L3 and N3, as observed in this study, are still unrevealed. Future works in that direction might shed some light on this. 

Although we speculate that a mixture of both L3 and N3 adjuvant in the same vaccine formulation might elicit both the humoral and cell-mediated arms of the immune response, it also remains one mystery that might be addressed in future studies. In any case, the standardization of the dose of such an L3+N3 mixture will be critical as both seem to be very active and can result in exaggerated immune response that might raise some safety concerns.

This study is critically important for demonstrating the differences in the mechanism of action of the differentially charged adjuvants in modulating the immune response elicited by the same vaccine candidate antigen. Combining our splenic and systemic immune response data here with the previously reported mucosal response [19,27,28,29] from the intra-nasal administration of lipid adjuvants can lead to a better understanding and improved design of customized vaccines. The immune responses generated proved that these adjuvants are not flushed out from the mucosal site of administration. This is one key concern for many intra-nasal vaccination strategies as they disturb the site of administration, causing local irritation and leading to them being flushed out, thus failing to provide a sufficient immune response [9,10].

## 5. Conclusions

In conclusion, we can highlight that the L3 anionic adjuvant elicited a stronger humoral response and is recommended to be used in combination with the vaccine antigen, which by itself fails to boost optimum humoral immunity. This can also be a very potent adjuvant to be used in designing customized vaccines for individuals with insufficient humoral response. On the other hand, the N3 cationic adjuvant elicited higher cell-mediated immunity and was a better choice of adjuvant to use with a vaccine-candidate antigen which failed to generate optimum cellular immunity.

These differentially charged lipid adjuvants can be potentially used in vaccinations to target populations with deficiencies in a single arm of the immune responses. A vast majority of the population has a weaker immune response due to cell-mediated or humoral immunity depending on their comorbidities, age, gender, etc. It is convenient to change the adjuvant accompanying the same vaccine antigen rather than looking for another antigen candidate for treating a particular disease. Especially in the context of the COVID-19 vaccine, we have learned that while the humoral response is important for sterilizing immunity, the presence of a sufficient T cell response is important in protecting against severe disease as antibody levels wane. Thus, the proper use and administration of different adjuvants may be the key to better protection against the influenza virus.

## Figures and Tables

**Figure 1 vaccines-12-00320-f001:**
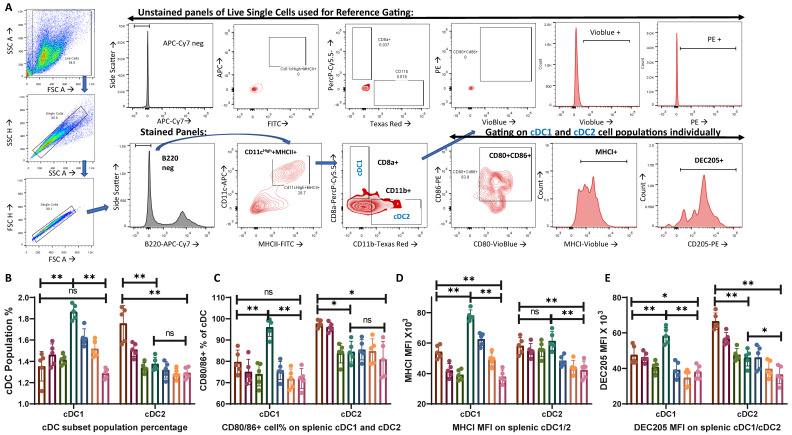
Analysis of population percentage and activation of conventional dendritic cell subsets: (**A**) FSC and SSC plots were used to display the live single-cell populations, and, from those, we gated out the B220 negative populations. They were then gated to determine MHCII+CD11c+ populations. Subsequently, they were gated to determine the CD8a+ and CD11b+ populations to obtain the B220−MHCII+CD11c+CD11b-CD8a+ cDC1 and B220−MHCII+CD11c+CD11b+CD8a− cDC2 population percentages. These two dendritic cell subsets were then individually assessed for CD80+CD86+, MHCI+, and DEC205+ cDC1 and cDC2. Unstained panels from the live single cells are shown in the upper row, which creates the reference gate for the stained samples below. (**B**) The bar diagram represents the population percentage of splenocytes for cDC1 and cDC2. (**C**) CD80+CD86+ cDC1 and cDC2 populations are represented in the bar diagram as the percentage of individual cDC1 and cDC2 populations. (**D**) Mean fluorescence intensity (MFI) of MHCI expression in cDC1 and cDC2 populations is represented here in the bar diagram. (**E**) DEC205 MFI expression within cDC1 and cDC2. The graphs are representative images derived from at least four independent experiments. Each round of experiments included five mice in each set: (* *p* < 0.05, ** *p* < 0.01, ns = non-significant). We followed the same controls, antibodies, machine settings, and gating strategies as our recently published work [44].

**Figure 2 vaccines-12-00320-f002:**
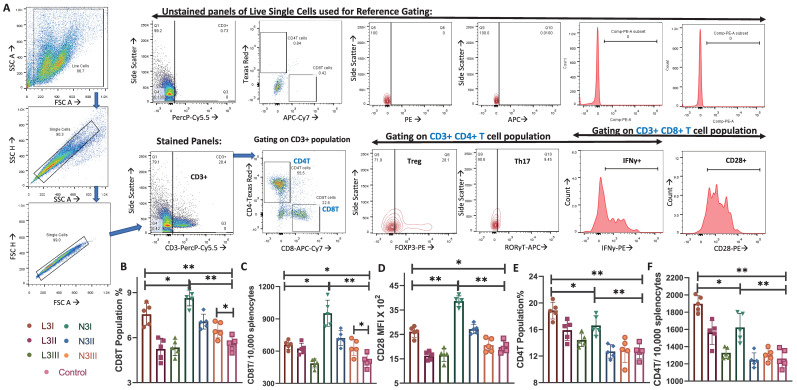
Cytotoxic CD8T cell and CD4T cell population dynamics and activation: (**A**) CD3+ cells were gated out from the live single-cell splenocyte suspension and were visualized using FSC and SSC plots. In the CD3+ gated population, CD4+ and CD8+ T cells were gated. The FOXP3, RORγT, and IFNγ expressions in CD3+CD4+ cells were analyzed. The CD3+CD8+ T cells were studied for IFNγ+ and CD28+ positive cells and MFI. Unstained gating (the upper row) was considered when drawing the respective gates for the stained samples (lower panel row). (**B**) Population percentage of CD3+CD8+ T cells out of the total splenocytes studied. (**C**) The absolute number of CD8T cells is shown for each group, normalized to 10,000 splenocytes. (**D**) The MFI of the activation and expression of the costimulatory marker of CD28. (**E**) The population percentage of CD3+CD4+ T cells. The percentage of CD4T cells shown here is on all splenocytes. (**F**) The absolute number of CD8T cells is shown for each group as normalized to 10,000 splenocytes. The graphs are representative images of at least four independent experiments. Each round of experiments had five mice in each set: (* *p* < 0.05, ** *p* < 0.01). We followed the same controls, antibodies, machine settings, and gating strategies as our recently published work [44].

**Figure 3 vaccines-12-00320-f003:**
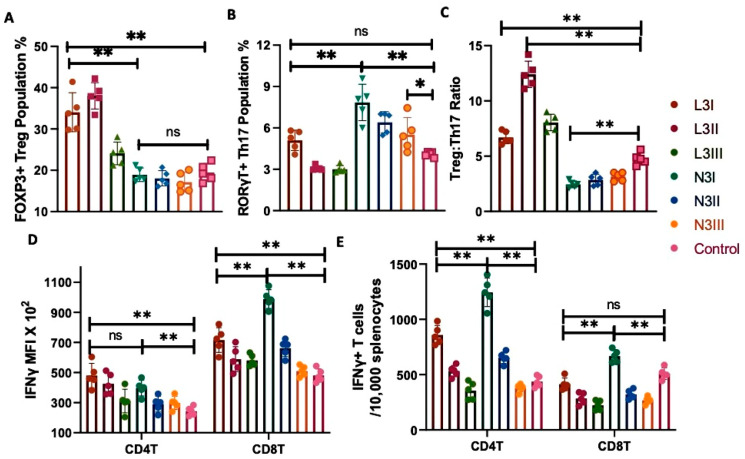
Treg and Th17 cell populations and IFNγ+ CD4T and IFNγ+ CD8T cell subsets: (**A**) The differentiation of CD4T cells into CD3+CD4+FoxP3+RORγT- regulatory T cells was significantly higher in the L3 groups whereas (**B**) the CD3+CD4+FoxP3-RORγT+ Th17 cell population is higher in N3 groups. (**C**) The ratio of Treg:Th17 is much higher in the L3 group, whereas a significantly lower ratio in the N3 groups was observed. (**D**) The MFI level of IFNγ+ expression within the T cell subsets is shown by the bar diagram. (**E**) The absolute numbers of IFNγ+ CD4T and IFNγ+ CD8T cells are provided in the bar diagram, normalized to 10,000 splenocytes. The graphs are representative images derived from at least four independent experiments. Each round of experiments had five mice in each set. (* *p* < 0.05, ** *p* < 0.01). We followed the same controls, antibodies, machine settings, and gating strategies as our recently published work [44].

**Figure 4 vaccines-12-00320-f004:**
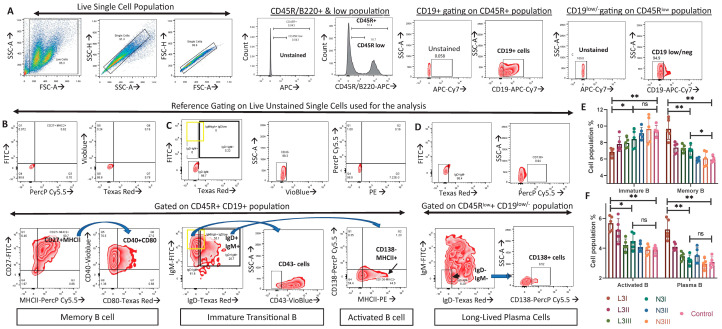
Population percentage of immature/transitional, memory, activated B, and plasma cells: (**A**) The B220^low^ and B220+ cell populations were gated out from live single splenocyte suspensions, keeping unstained samples as a reference for gating. CD19^low/neg^ cell populations were gated out from B220^low^ populations, and CD19+ cells were gated out from B220+ cells. (**B**) Within the B220+CD19+ cell populations, CD27+MHCII+ populations were gated, of which CD40+CD80+ populations were determined to obtain the B220/CD45R+CD19+CD27+MHCII+CD40+CD80+ memory B cell population. (**C**) IgM+IgD+ (thick black box) and IgM^high^+IgD^low/neg^ populations (yellow box) were gated on B220+CD19+ populations. From these, B220/CD45R+CD19+IgM^high^+IgD^low/neg^+CD43− immature/transitional B cell and B220/CD45R+CD19+IgM+IgD+MHCII+CD138- activated B cell population were determined. (**D**) From the B220^low^ CD19^low/neg^ population, IgM−IgD− cells were determined to finally obtain B220^low^ CD19^low/neg^ IgM-IgD−CD138+ plasma cells. (**E**) The bar diagram represents the immature/transitional B and memory B cell population percentages. (**F**) The bar diagram represents the activated B and plasma cells. The percentages exhibited on the y-axis of 4E and 4F represent the percentage within a total of 10,000 splenocytes analyzed. Hence, to calculate the absolute cell number, readers can use the conversion where 1% on the y-axis = 100 cells. Data in the graphs are the representative images derived from at least four independent experiments (* *p* < 0.05, ** *p* < 0.01). We followed the same controls, antibodies, machine settings, and gating strategies as our recently published work [44].

**Figure 5 vaccines-12-00320-f005:**
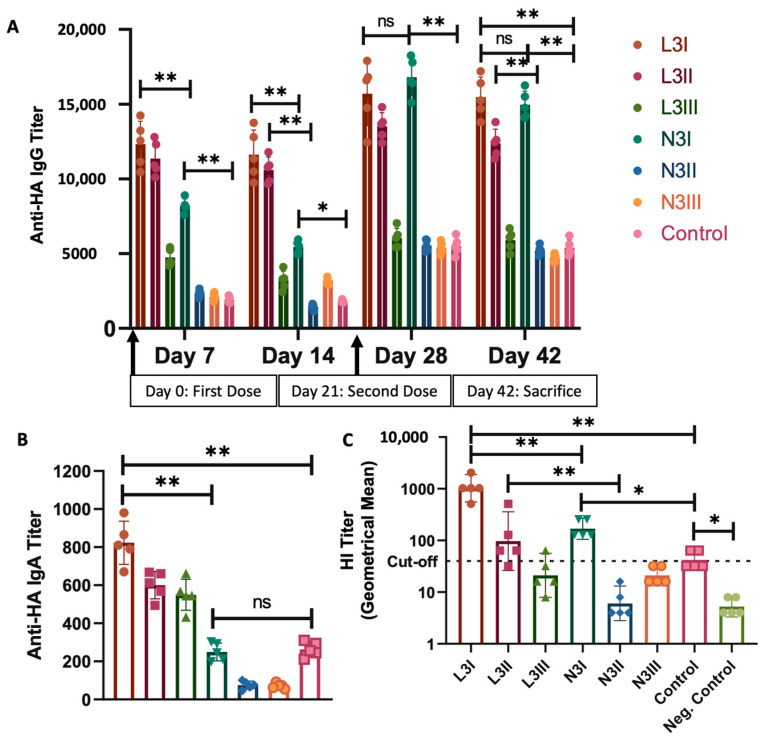
Higher expression of serum anti-HA IgG with enhanced neutralization capacity was observed in the N3 group: (**A**) Anti-HA IgG expression within the serum samples was determined at multiple timepoints post-first and -second dose of immunization by ELISA. (**B**) The anti-HA IgA titer was determined from the serum samples from each group on a HA-coated plate. (**C**) A hemagglutination Inhibition assay (HIA) was performed to determine the antibody response in the serum samples. The geometric means of the individual groups of the highest serum dilution capable of neutralizing the virus are plotted here, in log10 dilution factors represented by the *y*-axis. Titer > 40 was the cutoff for seroprotection titer and is marked on the graph. The graphs are representative images derived from at least four independent experiments. Each round of experiments had five mice in each set: (* *p* < 0.05, ** *p* < 0.01).

**Table 1 vaccines-12-00320-t001:** The characteristics of the seven experimental groups, each comprised of five mice. Dose signifies the dose received by a single mouse. The buffer used in the control is the same as the backbone solution used to prepare the adjuvant solution of L3 and N3. (Abbreviations: L3 = anionic lipid adjuvant, N3 = cationic lipid adjuvant, HA = hemagglutinin of influenza A/California/07/2009(H1N1)pdm).

Group Name	Antigen (HA) Dose	Adjuvant 2%	Sample Size
L3I	1 μg	L3 Anionic	5
L3II	0.1 μg	L3 Anionic	5
L3III	0.01 μg	L3 Anionic	5
N3I	1 μg	N3 Cationic	5
N3II	0.1 μg	N3 Cationic	5
N3III	0.01 μg	N3 Cationic	5
Control	1 μg	No Adjuvant/ Buffer	5

**Table 2 vaccines-12-00320-t002:** The antibody combinations used in flow cytometry.

Flow Cytometry Antibody Combinations	Phenotypes/Activation Marker	Antibodies Used (Catalogue No.) *	Refs.
Dendritic Cells		B220-APC-Vio770 (130-102-267) MHCII-FITC (130-112-386) CD11c-APC (130-119-762) CD11b-PE-Vio615 (130-113-807) CD8a-PerCP-Vio700 (130-128-616) CD80-VioBlue/PB (Biolegend: 104723) CD86-PE (130-123-724) MHCI-Vioblue (130-125-228) DEC205-PE (130-112-272)	[38,39]
B220-MHCII+CD11c+CD11b−CD8a+	Conventional dendritic cells 1 (cDC1)		
B220-MHCII+CD11c+CD11b+ CD8a-	Conventional dendritic cells 2 (cDC2)		
cDC1/cDC2+CD80+CD86	Co-stimulatory expression		
cDC1/cDC2+MHCI	MHC/co-stimulation marker		
cDC1/cDC2+DEC205	Activation/co-stimulation marker		
T cells:		CD3-PerCP-Vio700 (130-120-826) CD8-APC-Vio770 (130-123-280) CD4-PE-Vio615 (130-118-455) FOXP3-PE (130-111-678) RORgT-APC (130-124-035) CD28-PE (130-102-601) INFγ-PE (130-123-700)	[40,41]
CD3+CD8+	Cytotoxic T cells		
CD3+CD8+CD28	Co-stimulatory expression		
CD3+CD4+	Helper T cells		
CD3+CD4+FOXP3+	Treg subset of CD4T cells		
CD3+CD4+RORγT+	Th17 subset of CD4T cells		
B cells:		B220-APC (130-110-847) CD19-APC-Vio770 (130-123-572) MHCII-PerCP-Vio700 (130-112-391) MHCII-PE (130-112-387) IgM-FITC (130-095-906) IgD-BV605/TexasRed (Biolegend: 405727) CD138-PerCP-Cy5.5 (Biolegend: 142509) CD40-Vioblue/PB (Biolegend: 124625) CD80-PE-Vio615 (130-116-467) CD43-Vioblue (130-112-702) CD27-FITC (130-114-165)	[42,43]
B220^low^CD19 ^low/neg^ IgM-IgD-CD138+	Long-lived plasma cells		
B220/CD45R+CD19+IgM+IgD+MHCII+CD138-	Activated B cell		
B220/CD45R+CD19+IgM^high^+IgD^low/neg^+CD43-	Immature/Transitional B		
CD45R/B220+CD19+ CD27+MHC2+CD80+CD40+	Memory B cells		

* All antibodies were obtained from Miltenyi Biotec, unless otherwise mentioned.

**Table 3 vaccines-12-00320-t003:** Comparative summary of the two differentially charged adjuvants used in this study.

Parameters	L3 Anionic	N3 Cationic	Fig/Ref.
Dendritic cells	Increased cDC2%	Increase in cDC1%	Figure 1B
	Higher CD80/86+ cDC2	Higher CD80/86+ cDC1	Figure 1C
	Increased in MHCI+ cDC1 than control	Significantly higher MHCI+ cDC1 than cDC2	Figure 1D
	Increased DEC205 in cDC2	Increased DEC205 in cDC1	Figure 1E
T lymphocytes	Significantly lower CD8T population and CD28+CD8T than N3	Significantly high CD8T population and CD28+CD8T than L3	Figure 2B–D
	Significantly higher CD4T than N3 and control	Significantly lower than L3, higher than control	Figure 2E,F
	Treg: Significantly higher than N3 and control	Treg: No change compared to the control	Figure 3A
	Th17: No change than control	Th17: Significantly higher than L3 and control	Figure 3B
	Low IFNγ within CD8T, high within CD4T	High IFNγ within CD4T and CD8T	Figure 3D
B lymphocytes	Few immature B and high memory B cells	More immature B and low memory B cells	Figure 4E
	Significantly more activated B cells	No significant change compared to control	Figure 4F
	Significantly more plasma cells	No significant change compared to control	Figure 4F
Antibody titer	Higher anti-HA IgG compared to N3 and control.	Significantly higher anti-HA IgG than control	Figure 5A
	Significantly higher anti-HA IgA than N3.	Significantly lower anti-HA IgA than L3 and control	Figure 5B
HIA titer	Significantly higher than N3	Significantly lower than L3, higher than control	Figure 5C
Previous Report	24-fold increase in HIA titer compared to non-adjuvanted formulation	4-fold increase in HIA titer compared to non-adjuvanted formulation	Ref. [12]
	ELISPOT: 7-fold increase in T cell response as pg/mL of INF-γ over non-adjuvanted formulation	60-fold increase in T cell response as pg/mL of INF-γ over non-adjuvanted formulation	
Conclusion	Generated a stronger humoral immune response. Recommended formulation for: Individuals with insufficient humoral response Use with candidate vaccine antigen with suboptimal potential for humoral pathway activation	Generated stronger cellular Th1/Th17 immunity. Recommended formulation for: Individuals with insufficient cellular immunity Use with candidate vaccine antigen with suboptimal potential for activation of cell-mediated immunity	

## Data Availability

All the data used and/or analyzed during the current study are available in the main manuscript.

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
