# Peer review of "Differential Immune Response Patterns Induced by Anionic and Cationic Lipid Adjuvants in Intranasal Anti-Influenza Immunization"

_vaccines, 2024, doi:10.3390/vaccines12030320_

Round 1

Reviewer 1 Report

Comments and Suggestions for Authors

The manuscript introduces an important animal model for testing the suitable adjuvants for vaccination against respiratory diseases.  Based on adequate and important issue, appropriate methodology and well-demonstrated results, I recommend the manuscript for publication.

Author Response

REVIEW 1

Open Review

Quality of English Language

( ) I am not qualified to assess the quality of English in this paper
( ) English very difficult to understand/incomprehensible
( ) Extensive editing of English language required
( ) Moderate editing of English language required
( ) Minor editing of English language required
(x) English language fine. No issues detected

Yes

Can be improved

Must be improved

Not applicable

Does the introduction provide sufficient background and include all relevant references?

(x)

( )

( )

( )

Are all the cited references relevant to the research?

(x)

( )

( )

( )

Is the research design appropriate?

(x)

( )

( )

( )

Are the methods adequately described?

(x)

( )

( )

( )

Are the results clearly presented?

( )

( )

( )

( )

Are the conclusions supported by the results?

(x)

( )

( )

( )

Comments and Suggestions for Authors

The manuscript introduces an important animal model for testing the suitable adjuvants for vaccination against respiratory diseases.  Based on adequate and important issue, appropriate methodology and well-demonstrated results, I recommend the manuscript for publication.

Reply: We would like to thank the reviewer for his/her positive response to our manuscript. We are indeed thankful to him/her to find out time from his/her busy schedule to review our manuscript.

Reviewer 2 Report

Comments and Suggestions for Authors

Major Comments:

Sengupta and colleagues have examined the impact of adjuvant formulation using cationic and anionic lipids on the immune response to immunization with influenza HA protein.  I think the idea of this study is excellent, and that it is a critically under-studied area of immunology/vaccinology.  The manuscript has some very minor weaknesses.  The authors have looked at cellular responses in the spleen rather than in LN draining the respiratory tract.  The Figure 1 and Figure 4 need to be reformatted so that the dot plots are larger.  Figure 4 should be moved down in the text so that it appears after its first mention.  To be clear these are extremely minor criticisms.  I’m very enthusiastic about the manuscript.  I hope that the authors are able to focus in on the draining secondary lymphoid tissues in the near future because I expect the results in that compartment would be even more exciting that what they report from the spleen. 

Minor Comments:

1.       Line 41 – revise to “…have been moving towards…”

2.       I would insert a space between the reference number and the last letter of the word preceding the reference—just a suggestion and my personal preference rather than a criticism.  Either way, you should be consistent.  See line 66.

3.       Line 288 – revise “CD4T” to “CD4+ T”

Author Response

REVIEW 2

Open Review

Quality of English Language

( ) I am not qualified to assess the quality of English in this paper
( ) English very difficult to understand/incomprehensible
( ) Extensive editing of English language required
( ) Moderate editing of English language required
( ) Minor editing of English language required
(x) English language fine. No issues detected

Yes

Can be improved

Must be improved

Not applicable

Does the introduction provide sufficient background and include all relevant references?

(x)

( )

( )

( )

Are all the cited references relevant to the research?

(x)

( )

( )

( )

Is the research design appropriate?

(x)

( )

( )

( )

Are the methods adequately described?

(x)

( )

( )

( )

Are the results clearly presented?

(x)

( )

( )

( )

Are the conclusions supported by the results?

(x)

( )

( )

( )

Comments and Suggestions for Authors

Major Comments:

Sengupta and colleagues have examined the impact of adjuvant formulation using cationic and anionic lipids on the immune response to immunization with influenza HA protein.  I think the idea of this study is excellent, and that it is a critically under-studied area of immunology/vaccinology.  The manuscript has some very minor weaknesses.  The authors have looked at cellular responses in the spleen rather than in LN draining the respiratory tract.  The Figure 1 and Figure 4 need to be reformatted so that the dot plots are larger.  Figure 4 should be moved down in the text so that it appears after its first mention.  To be clear these are extremely minor criticisms.  I’m very enthusiastic about the manuscript.  I hope that the authors are able to focus in on the draining secondary lymphoid tissues in the near future because I expect the results in that compartment would be even more exciting that what they report from the spleen. 

Reply: We would like to thank the reviewer for his/her positive response to our manuscript. We are indeed thankful to him/her to find out time from his/her busy schedule to review our manuscript.

It is indeed a great suggestion to expand our study to focus on the draining secondary lymphoid tissues in future. We would keep that in mind while designing future works with our adjuvants. Regarding the Figure 1 and 4 the dot plots hopefully will be much better in the final version when/if it is published. We would be happy to provide high resolution figures to the Journal before it goes to press.

Minor Comments:

  1. Line 41 – revise to “…have been moving towards…”

Reply: We have edited as suggested. Line no: 40

  1. I would insert a space between the reference number and the last letter of the word preceding the reference—just a suggestion and my personal preference rather than a criticism. Either way, you should be consistent.  See line 66.

Reply: We have edited as suggested. Line no: 65

  1. Line 288 – revise “CD4T” to “CD4+ T”

Reply: We have edited as suggested. Line No: 288

Reviewer 3 Report

Comments and Suggestions for Authors

Sengupta et al. conducted a side-by-side comparison of differentially charged lipid adjuvants in stimulation of systemic immune responses following intranasal immunization of influenza HA antigens in murine models. Both adjuvants (L3 anionic, N3 cationic) have been extensively tested in preclinical animal models following intranasal immunization. The current study focused on evaluation of their differential potentiation of influenza HA-induced systemic humoral and cellular immune responses. The major findings were L3 adjuvant preferentially stimulated cDC2, Treg, B cell, anti-HA antibody responses, while N3 adjuvant preferentially stimulated cDC1, Th17, CD8+ T cell responses, which were in line with prior reports. I have below comments to help to improve the manuscript.

1.      It’s recommended to add a side-by-side comparison of the two adjuvants including below information: particle size, surface charge, composition, etc.

2.      It’s recommended to move gating strategies in Fig.1, 2 and 4 to supplementary document and only keep main data in the main text.

3.      Adjuvant preparation (lines 81-89) can be moved to methods.

4.      Lines 200-203 are not needed since it’s typical to evaluate serum antibody responses 21 days after immunization in preclinical studies.

5.      Lines 309-312 can be moved to figure legends. Similar to lines 352-355 and other similar ones.

6.      No absolute cell numbers were presented in Fig.4E and 4F, while lines 416-417 mentioned absolute cell numbers.

7.      Lines 437-438 should be modified to ‘N3I treatment reduced immature B cells …’ (due to reduced values when compared to control group).

8.      Values in control group of Fig.3E, 5A, and 5B were higher than that in immunized groups, why?

9.      Line 558-559 need to cite a reference.

10.  Grammatical errors need to be corrected throughout.

Comments on the Quality of English Language

English needs to be improved. 

Author Response

REVIEW 3

Open Review

Quality of English Language

( ) I am not qualified to assess the quality of English in this paper
( ) English very difficult to understand/incomprehensible
(x) Extensive editing of English language required
( ) Moderate editing of English language required
( ) Minor editing of English language required
( ) English language fine. No issues detected

Yes

Can be improved

Must be improved

Not applicable

Does the introduction provide sufficient background and include all relevant references?

( )

(x)

( )

( )

Are all the cited references relevant to the research?

(x)

( )

( )

( )

Is the research design appropriate?

( )

(x)

( )

( )

Are the methods adequately described?

(x)

( )

( )

( )

Are the results clearly presented?

( )

(x)

( )

( )

Are the conclusions supported by the results?

( )

(x)

( )

( )

Comments and Suggestions for Authors

Sengupta et al. conducted a side-by-side comparison of differentially charged lipid adjuvants in stimulation of systemic immune responses following intranasal immunization of influenza HA antigens in murine models. Both adjuvants (L3 anionic, N3 cationic) have been extensively tested in preclinical animal models following intranasal immunization. The current study focused on evaluation of their differential potentiation of influenza HA-induced systemic humoral and cellular immune responses. The major findings were L3 adjuvant preferentially stimulated cDC2, Treg, B cell, anti-HA antibody responses, while N3 adjuvant preferentially stimulated cDC1, Th17, CD8+ T cell responses, which were in line with prior reports. I have below comments to help to improve the manuscript.

  1. It’s recommended to add a side-by-side comparison of the two adjuvants including below information: particle size, surface charge, composition, etc.

Reply: We have tried our best to address the issue highlighted by the reviewer here. Unfortunately, despite our best effort, we are unable to provide every details as asked here. These adjuvants are proprietary to Eurocine Vaccines AB, Sweden, and have been used in multiple studies before. We also received the adjuvant from the company and mentioned it in the line 129-130. As we have not prepared it ourselves the small details about its characterization is not known to us. According to the company policy, much of the information is restricted from being published as well, and we have not received those from them.  We tried to provide as much information as we possibly get about these adjuvants from the previous dozens of publications, and also from the patents and clinical trials of these adjuvants.

Some of the sources cited in the reference regarding the clinical trials, thesis, patents and research articles with these adjuvants are listed below for the ready reference:

Clinical Trials:

  1. https://classic.clinicaltrials.gov/ct2/show/NCT03437304
  2. https://classic.clinicaltrials.gov/ct2/show/NCT02998996

PhD Thesis:

Patents:

  1. CA2258017C - Immunstimulating lipid formulation - Google Patents [Internet]. [cited 2023 Jul 4]. Available from: https://patents.google.com/patent/CA2258017C/en?q=(eurocine+adjuvant+influenza)&assignee=Eurocine+Vaccines+Ab
  2. AU2011310090B2 - Improved vaccine compositions - Google Patents [Internet]. [cited 2023 Jul 4]. Available from: https://patents.google.com/patent/AU2011310090B2/en?q=(eurocine+adjuvant+influenza)&assignee=Eurocine+Vaccines+Ab
  3. WO2015110659A1 - Methods of immunization with a vaccine inducing a humoral immune response and with a vaccine inducing a cellular immune response - Google Patents [Internet]. [cited 2023 Jul 4]. Available from: https://patents.google.com/patent/WO2015110659A1/en?q=(eurocine+adjuvant+influenza)&oq=eurocine+adjuvant+influenza
  4. JP2019112448A - Vaccine composition for naive subjects - Google Patents [Internet]. [cited 2023 Jul 4]. Available from: https://patents.google.com/patent/JP2019112448A/en?q=(eurocine+adjuvant+influenza)&assignee=Eurocine+Vaccines+Ab
  5. US20150306205A1 - Vaccine composition for naive subjects - Google Patents [Internet]. [cited 2023 Jul 4]. Available from: https://patents.google.com/patent/US20150306205A1/en

Few Publications samples using these adjuvants:

Intranasally administered Endocine™ formulated 2009 pandemic influenza H1N1 vaccine induces broad specific antibody responses and confers protection in ferrets.
Maltais AK, Stittelaar KJ, Veldhuis Kroeze EJ, van Amerongen G, Dijkshoorn ML, Krestin GP, Hinkula J, Arwidsson H, Lindberg A, Osterhaus AD. Vaccine. 2014 May 30;32(26):3307-15. doi: 10.1016/j.vaccine.2014.03.061. Epub 2014 Mar 30.

Endocine™, N3OA and N3OASq; three mucosal adjuvants that enhance the immune response to nasal influenza vaccination.
Falkeborn T, Bråve A, Larsson M, Akerlind B, Schröder U, Hinkula J. PLoS One. 2013 Aug 8;8(8):e70527. doi: 10.1371/journal.pone.0070527. eCollection 2013

The Eurocine L3 adjuvants with subunit influenza antigens induce protective immunity in mice after intranasal vaccination.
Pernilla Petersson, Mona Hedenskog, Denise Alves, Mia Brytting, Ulf Schröder, Annika Linde, Åke Lundkvist. Vaccine 28 (2010) 6491-6497

Safety and immunogenicity, after nasal application of HIV-1 DNA gagp37 plasmid vaccine in young mice.
Jorma Hinkula, Marie Hagbom, Britta Wahren, Ulf Schröder. Vaccine 26 (2008) 5101-5106

Intranasal immunization of young mice with a multigene HIV-1 vaccine in combination with the N3 adjuvant induces mucosal and systemic immune responses.
Andreas Bråve, David Hallengärd, Ulf Schöder, Pontus Blomberg, Britta Wahren and Jorma Hinkula. Vaccine 26 (2008) 5075-5078

DNA–VLP prime–boost intra-nasal immunization induces cellular and humoral anti-HIV-1 systemic and mucosalimmunity with cross-clade neutralizing activity.
L. Buonaguro, C. Devito, M.L. Tornesello, U. Schröder, B. Wahren, J. Hinkula and F.M. Buonaguro Vaccine 25 (2007) 5968-5977

A novel DNA adjuvant, N3, enhances mucosal and systemic immune responses induced by HIV-1 DNA and peptide immunizations.
Jorma Hinkula, Claudia Devito, Bartek Zuber, Reinhold Benthin, Denise Ferreira, Britta Wahren, Ulf Schröder. Vaccine 24 (2006) 4494-4497

Nasal boost with adjuvanted heat killed BCG or arabinomannan-protein conjugate improves primary BCG-induced protection in C57BL/6 mice.
M. Haile, B. Hamasur, T. Jaxmar, D. Gavier-Widen, M.A. Chambers, B. Sanchez, U. Schröder, Källenius, S.B. Svenson, A. Pawlowski. Tuberculosis (2005) 85, 107-114

Intranasal HIV-1-gp160-DNA/gp41 peptide prime-boost immunization regimen of mice results in longterm HIV-1 neutralizing humoral, mucosal and systemic immunity.
C. Devito, B. Zuber, U. Schröder, R. Benthin, K. Okuda, K. Broliden, B. Wahren and J. Hinkula. J.Immunology (2004) 173: 7078-7089

Immunization with heat-killed Mycobacterium bovis bacille Calmette–Guerin (BCG) in EurocineTM L3 adjuvant protects against tuberculosis.
M. Haile, U. Schröder, B. Hamasur, A. Pawlowski, T. Jaxmar, G. Källenius, S.B. Svenson. Vaccine 22 (2004) 1498-1508

The details of the company manufacturing these adjuvants:

https://www.eurocine-vaccines.com/the-portfolio/

  1. It’s recommended to move gating strategies in Fig.1, 2 and 4 to supplementary document and only keep main data in the main text.

Reply: Thanks for the suggestion. We are happy to move in the gating strategies to the supplementary. However, we find it is easier for the readers to understand exactly which subset of cells we are talking about in the main text of the manuscript if they can find it in the main figure. For example, in Fig 4, we mentioned the cell subsets by marking them as ‘Thick box’ or ‘Yellow marked box’ in the text so as to make it easier for the readers to understand. If it is absolutely necessary to get the work published, we will be happy to move the gating strategies to the supplementary in the next submission. For now, we are keeping it same and free from having any supplementary document for this manuscript.

  1. Adjuvant preparation (lines 81-89) can be moved to methods.

Reply: Thanks for the suggestion. We moved the whole section to the methods as suggested in the line no: 175-183 (yellow shaded).

  1. Lines 200-203 are not needed since it’s typical to evaluate serum antibody responses 21 days after immunization in preclinical studies.

Reply: Thanks for the suggestion. We mentioned about it for the readers who are not directly involved in the serological studies. However, we have deleted those sentences in this revised submission as suggested here. (Line no 198-203 Strikethrough in yellow-shaded text)

  1. Lines 309-312 can be moved to figure legends. Similar to lines 352-355 and other similar ones.

Reply: They are moved to the Figure legends as suggested. We meant to provide them in the figure legend only but unfortunately due to some mistake in our part during formatting they got into the main text part. Now we have edited it correctly. Thanks for noticing this. Line No: 308-310, 351-354, 383-386, 414-417 in the figure legends. (yellow shaded).

  1. No absolute cell numbers were presented in Fig.4E and 4F, while lines 416-417 mentioned absolute cell numbers.

Reply: Sorry for not making ourselves clear in our previous submission. We provided that sentence for the understanding for the readers about the absolute number then they can calculate themselves if they require that. For Example, in Fig 4E and 4F, we provided population percentages in Y axis. Sometimes, it is confusing because percentages ‘of what’ is not clear. For example percentage of Immature B cells in 4E can be percentage of Lymphocytes or percentage of Live cells or something else. Hence, we mention about the absolute number calculation that can be deduced from it so as to know exact how many numbers of cells we are talking here. The graphs in Fig 4E and 4F will be exactly same if we replace the Y axis with 1% of Y axis is 100 cells.  However, we understand that sentence might be confusing to many as the reviewer rightly pointed out. So we reframed the sentence in Line no: 413-414.

  1. Lines 437-438 should be modified to ‘N3I treatment reduced immature B cells …’ (due to reduced values when compared to control group).

Reply: Sorry for not making ourselves clear in our previous submission. We indeed mentioned that both L3 and N3 treatment reduced the immature B cell (as compared to the control) just one line before that (Line No:440). However the particular line that reviewer pointed out is the comparison between L3 and N3 adjuvant, where N3I group has significantly higher immature B than L3I (Line No: 442). Hence, we are keeping the same sentences and not changing them as suggested.

  1. Values in control group of Fig.3E, 5A, and 5B were higher than that in immunized groups, why?

Reply: Thank you for pointing this out. We are sorry that we have not make ourselves clear in the previous submission. The control itself contains the antigen dose (without the adjuvants). So it is expected to have some immune response atleast. The control bar which is the last one of the group is significantly lower or have no significant change than highest dose of the adjuvant-antigen formulation (L3I or N3I). However, L3II/ L3III and N3II/N3III groups might have lower response than the control in some cases like as reviewer said as the antigen dose is lower in them than the control. As mentioned in the Table 1, L3I, N3I and the control has the same highest and similar amount of antigen, whereas L3II/L3III and N3II/N3III has lower antigen doses. Hence the lower antigen results in the lower immunization response even with the same dose of adjuvant. If compared with ‘L3I vs control’ or ‘N3I vs control’, all of which have the same antigen dose then the values of the control are not higher than the immunized groups. L3I/N3I immunization response is either higher or have no significant change with the control in all these figures.

  1. Line 558-559 need to cite a reference.

Reply: We added the reference. Reference No: 62 in this revised version it is in line no: 562.

  1. Grammatical errors need to be corrected throughout.

Reply: Thanks for the suggestion. We tried our best to improve the English language editing. We have edited the manuscript through paid English language editing service Editage. We also checked ourselves the English language throughout in this revised submission.

Comments on the Quality of English Language

English needs to be improved. 

Reply: Thanks for the suggestion. We tried our best to improve the English language editing. We have edited the manuscript through paid English language editing service Editage. We also checked ourselves the English language throughout in this revised submission.

Reviewer 4 Report

Comments and Suggestions for Authors

Sengupta et al. evaluated effects of lipid adjuvant charge on immunization responses. They showed that the cationic adjuvant (N3) increases cDC1 with MHC class I, CD80-CD86 higher expression, CD8+ T cells and Th17, and that the anionic adjuvant (L3) increases cDC2 with MHC class II, DEC205 higher expression, CD4+ T cells and Treg. Further, they found that anionic adjuvant (L3) groups showed higher antigen-specific IgG and IgA induction. The manuscript is well written and the topic is interesting.

I have some comments on the manuscript.

The results shown are interesting, but the results are totally phenomenological and there is no description concerning the underlying mechanisms. Do authors have some idea about how difference of adjuvant charge affects immune responses? It is desirable to show some speculated mechanisms in Discussion section.

The adjuvant preparation method is important. Therefore, it is better to describe the protocol briefly in this manuscript again (lines 177-180).

It is interesting the immunological effects of the mixture of L3 and N3 adjuvants. Hopefully, it may induce both humoral and cellular immunity. What is authors’ idea?

Table 3 seems somewhat wordy. It should be improved.

Minor comments:

Lines 309-312, 352-355, 385-388, 415-420, 456-457. They are similar sentences. Check the sentences again. They should be in figure legends, not in text.

Line 509. Correct the style of Reference #12.

Format of References is not correct. Check it carefully.

Comments on the Quality of English Language

Minor editing of English language required.

Line 188. “affects” reads “effects”.

Author Response

REVIEW 4

Open Review

Quality of English Language

( ) I am not qualified to assess the quality of English in this paper
( ) English very difficult to understand/incomprehensible
( ) Extensive editing of English language required
( ) Moderate editing of English language required
(x) Minor editing of English language required
( ) English language fine. No issues detected

Yes

Can be improved

Must be improved

Not applicable

Does the introduction provide sufficient background and include all relevant references?

(x)

( )

( )

( )

Are all the cited references relevant to the research?

( )

(x)

( )

( )

Is the research design appropriate?

(x)

( )

( )

( )

Are the methods adequately described?

( )

(x)

( )

( )

Are the results clearly presented?

(x)

( )

( )

( )

Are the conclusions supported by the results?

(x)

( )

( )

( )

Comments and Suggestions for Authors

Sengupta et al. evaluated effects of lipid adjuvant charge on immunization responses. They showed that the cationic adjuvant (N3) increases cDC1 with MHC class I, CD80-CD86 higher expression, CD8+ T cells and Th17, and that the anionic adjuvant (L3) increases cDC2 with MHC class II, DEC205 higher expression, CD4+ T cells and Treg. Further, they found that anionic adjuvant (L3) groups showed higher antigen-specific IgG and IgA induction. The manuscript is well written and the topic is interesting.

I have some comments on the manuscript.

The results shown are interesting, but the results are totally phenomenological and there is no description concerning the underlying mechanisms. Do authors have some idea about how difference of adjuvant charge affects immune responses? It is desirable to show some speculated mechanisms in Discussion section.

Reply: It is indeed an excellent suggestion. We included couple of paragraphs in this context from line no: 578-598 with some important references. The reviewer however is absolutely correct that we did not decipher the underlying mechanism that might to lead to the effect that we are finding using these adjuvants. We will keep a note of that and will definitely try to expand our study to find the mechanism in future. In this manuscript we mainly focused on reporting the alternative immune response generated by these adjuvants.

The adjuvant preparation method is important. Therefore, it is better to describe the protocol briefly in this manuscript again (lines 177-180).

Reply: We have tried our best to address the issue highlighted by the reviewer here. Unfortunately, despite our best effort, we are unable to provide every details as asked here. These adjuvants are proprietary to Eurocine Vaccines AB, Sweden, and have been used in multiple studies before. We also received the adjuvant from the company and mentioned it in the line 129-130. As we have not prepared it ourselves the small details about its characterization is not known to us. According to the company policy, much of the information is restricted from being published as well, and we have not received those from them.  We tried to provide as much information as we possibly get about these adjuvants from the previous dozens of publications, and also from the patents and clinical trials of these adjuvants. We tried to explain the protocol briefly in line nos: 175-183.

Some of the sources cited in the reference regarding the clinical trials, thesis, patents and research articles with these adjuvants are listed below for the ready reference:

Clinical Trials:

  1. https://classic.clinicaltrials.gov/ct2/show/NCT03437304
  2. https://classic.clinicaltrials.gov/ct2/show/NCT02998996

PhD Thesis:

Patents:

  1. CA2258017C - Immunstimulating lipid formulation - Google Patents [Internet]. [cited 2023 Jul 4]. Available from: https://patents.google.com/patent/CA2258017C/en?q=(eurocine+adjuvant+influenza)&assignee=Eurocine+Vaccines+Ab
  2. AU2011310090B2 - Improved vaccine compositions - Google Patents [Internet]. [cited 2023 Jul 4]. Available from: https://patents.google.com/patent/AU2011310090B2/en?q=(eurocine+adjuvant+influenza)&assignee=Eurocine+Vaccines+Ab
  3. WO2015110659A1 - Methods of immunization with a vaccine inducing a humoral immune response and with a vaccine inducing a cellular immune response - Google Patents [Internet]. [cited 2023 Jul 4]. Available from: https://patents.google.com/patent/WO2015110659A1/en?q=(eurocine+adjuvant+influenza)&oq=eurocine+adjuvant+influenza
  4. JP2019112448A - Vaccine composition for naive subjects - Google Patents [Internet]. [cited 2023 Jul 4]. Available from: https://patents.google.com/patent/JP2019112448A/en?q=(eurocine+adjuvant+influenza)&assignee=Eurocine+Vaccines+Ab
  5. US20150306205A1 - Vaccine composition for naive subjects - Google Patents [Internet]. [cited 2023 Jul 4]. Available from: https://patents.google.com/patent/US20150306205A1/en

Few Publications samples using these adjuvants:

Intranasally administered Endocine™ formulated 2009 pandemic influenza H1N1 vaccine induces broad specific antibody responses and confers protection in ferrets.
Maltais AK, Stittelaar KJ, Veldhuis Kroeze EJ, van Amerongen G, Dijkshoorn ML, Krestin GP, Hinkula J, Arwidsson H, Lindberg A, Osterhaus AD. Vaccine. 2014 May 30;32(26):3307-15. doi: 10.1016/j.vaccine.2014.03.061. Epub 2014 Mar 30.

Endocine™, N3OA and N3OASq; three mucosal adjuvants that enhance the immune response to nasal influenza vaccination.
Falkeborn T, Bråve A, Larsson M, Akerlind B, Schröder U, Hinkula J. PLoS One. 2013 Aug 8;8(8):e70527. doi: 10.1371/journal.pone.0070527. eCollection 2013

The Eurocine L3 adjuvants with subunit influenza antigens induce protective immunity in mice after intranasal vaccination.
Pernilla Petersson, Mona Hedenskog, Denise Alves, Mia Brytting, Ulf Schröder, Annika Linde, Åke Lundkvist. Vaccine 28 (2010) 6491-6497

Safety and immunogenicity, after nasal application of HIV-1 DNA gagp37 plasmid vaccine in young mice.
Jorma Hinkula, Marie Hagbom, Britta Wahren, Ulf Schröder. Vaccine 26 (2008) 5101-5106

Intranasal immunization of young mice with a multigene HIV-1 vaccine in combination with the N3 adjuvant induces mucosal and systemic immune responses.
Andreas Bråve, David Hallengärd, Ulf Schöder, Pontus Blomberg, Britta Wahren and Jorma Hinkula. Vaccine 26 (2008) 5075-5078

DNA–VLP prime–boost intra-nasal immunization induces cellular and humoral anti-HIV-1 systemic and mucosalimmunity with cross-clade neutralizing activity.
L. Buonaguro, C. Devito, M.L. Tornesello, U. Schröder, B. Wahren, J. Hinkula and F.M. Buonaguro Vaccine 25 (2007) 5968-5977

A novel DNA adjuvant, N3, enhances mucosal and systemic immune responses induced by HIV-1 DNA and peptide immunizations.
Jorma Hinkula, Claudia Devito, Bartek Zuber, Reinhold Benthin, Denise Ferreira, Britta Wahren, Ulf Schröder. Vaccine 24 (2006) 4494-4497

Nasal boost with adjuvanted heat killed BCG or arabinomannan-protein conjugate improves primary BCG-induced protection in C57BL/6 mice.
M. Haile, B. Hamasur, T. Jaxmar, D. Gavier-Widen, M.A. Chambers, B. Sanchez, U. Schröder, Källenius, S.B. Svenson, A. Pawlowski. Tuberculosis (2005) 85, 107-114

Intranasal HIV-1-gp160-DNA/gp41 peptide prime-boost immunization regimen of mice results in longterm HIV-1 neutralizing humoral, mucosal and systemic immunity.
C. Devito, B. Zuber, U. Schröder, R. Benthin, K. Okuda, K. Broliden, B. Wahren and J. Hinkula. J.Immunology (2004) 173: 7078-7089

Immunization with heat-killed Mycobacterium bovis bacille Calmette–Guerin (BCG) in EurocineTM L3 adjuvant protects against tuberculosis.
M. Haile, U. Schröder, B. Hamasur, A. Pawlowski, T. Jaxmar, G. Källenius, S.B. Svenson. Vaccine 22 (2004) 1498-1508

The details of the company manufacturing these adjuvants:

https://www.eurocine-vaccines.com/the-portfolio/

It is interesting the immunological effects of the mixture of L3 and N3 adjuvants. Hopefully, it may induce both humoral and cellular immunity. What is authors’ idea?

Table 3 seems somewhat wordy. It should be improved.

Reply: Thank you for the suggestion. We tried to drop the words wherever possible in the table to make it more concise and easier to read for the readers. The table does not contain any new information that is not in the text. But we think a concise table will highlight all the major findings that are hiding in different parts of the text. Therefore we think having this table 3 will be helpful for the readers.

Minor comments:

Lines 309-312, 352-355, 385-388, 415-420, 456-457. They are similar sentences. Check the sentences again. They should be in figure legends, not in text.

Reply: They are moved to the Figure legends as suggested. We meant to provide them in the figure legend only but unfortunately due to some mistake in our part during formatting they got into the main text part. Now we have edited it correctly. Thanks for noticing this. Thanks for noticing this. Line No: 308-310, 351-354, 383-386, 414-417 in the figure legends. (yellow shaded).

Line 509. Correct the style of Reference #12.

Format of References is not correct. Check it carefully.

Reply: We tried to correct all the reference styles. However as stated before there are few references (for example: reference 11 to 15) where we provided the links to the patents that are out in the database for these adjuvants. For them the reference style does not match with others as they are weblinks from the database with the patent numbers mentioned.

Comments on the Quality of English Language

Minor editing of English language required.

Line 188. “affects” reads “effects”.

Reply: We have sent this for paid professional English editing service EDitage. We also checked ourselves the English language before this submission. We hope the quality of English in this version achived publication quality. We change the Line 188 error that Reviewer mentioned (Line no: 188).

Round 2

Reviewer 3 Report

Comments and Suggestions for Authors

Authors addressed most of my comments. Regarding ‘N3I treatment reduced immature B cells …’. Authors can simply change to ‘N3I group showed significantly higher levels of immature B cells …’. Using ‘induced’ is very confusing.

Comments on the Quality of English Language

Minor editing is needed.

Author Response

The changes in this version:

  1. Reviewer 3: The changes suggested is added in Line 441-442 and we framed the sentence as the reviewer 3 suggested. Thanks for the suggestion.

Reviewer 4 Report

Comments and Suggestions for Authors

Authors replied to reviewers’ comments in details. The manuscript was well improved after revision.

I just comment a minor point.

Authors stated that they changed the Line 188 error that reviewer mentioned (Line no:188) in their reply. However, Line 188 was not changed in revised version. “affects” is a verb and “effects” is a noun. I think that “effects” is correct in line 188. 

Author Response

The changes in this version:

  1. Reviewer 4: Sorry to miss the typo in Line 188. Now we changed the ‘affects’ to ‘effects’ as suggested by the reviewer.

We would like to thank all the reviewers and the editor(s) of the manuscript for their comments and suggestions to improve the manuscript.